# Study on Temperature-Dependent Properties and Fire Resistance of Metakaolin-Based Geopolymer Foams

**DOI:** 10.3390/polym12122994

**Published:** 2020-12-15

**Authors:** Van Su Le, Petr Louda, Huu Nam Tran, Phu Dong Nguyen, Totka Bakalova, Katarzyna Ewa Buczkowska, Iva Dufkova

**Affiliations:** 1Department of Material Science, Faculty of Mechanical Engineering, Technical University of Liberec, Studentska 2, 461 17 Liberec, Czech Republic; petr.louda@tul.cz (P.L.); totka.bakalova@tul.cz (T.B.); katarzyna.ewa.buczkowska@tul.cz (K.E.B.); iva.dufkova@tul.cz (I.D.); 2Department of Applied Mechanics, Faculty of Mechanical Engineering, Technical University of Liberec, Studentska 2, 461 17 Liberec, Czech Republic; thnam.hut@gmail.com; 3Department of Vehicles and Engines, Faculty of Mechanical Engineering, Technical University of Liberec, Studentska 2, 461 17 Liberec, Czech Republic; nguyenphudong89@gmail.com; 4Department of Materials Technology and Production Systems, Faculty of Mechanical Engineering, Lodz University of Technology, Stefanowskiego 1/15, 90-924 Lodz, Poland

**Keywords:** geopolymer foam, basalt fiber, high temperature, fire resistance

## Abstract

This paper presents temperature-dependent properties and fire resistance of geopolymer foams made of ground basalt fibers, aluminum foaming agents, and potassium-activated metakaolin-based geopolymers. Temperature-dependent properties of basalt-reinforced geopolymer foams (BGFs) were investigated by a series of measurements, including apparent density, water absorption, mass loss, drying shrinkage, compressive and flexural strengths, XRD, and SEM. Results showed that the apparent density and drying shrinkage of the BGFs increase with increasing the treated temperature from 400 to 1200 °C. Below 600 °C the mass loss is enhanced while the water absorption is reduced and they both vary slightly between 600 and 1000 °C. Above 1000 °C the mass loss is decreased rapidly, whereas the water absorption is increased. The compressive and flexural strengths of the BGFs with high fiber content are improved significantly at temperatures over 600 °C and achieved the maximum at 1200 °C. The BGF with high fiber loading at 1200 °C exhibited a substantial increase in compressive strength by 108% and flexural strength by 116% compared to that at room temperature. The enhancement in the BGF strengths at high temperatures is attributed to the development of crystalline phases and structural densification. Therefore, the BGFs with high fiber loading have extraordinary mechanical stability at high temperatures. The fire resistance of wood and steel plates has been considerably improved after coating a BGF layer on their surface. The coated BGF remained its structural integrity without any considerable macroscopic damage after fire resistance test. The longest fire-resistant times for the wood and steel plates were 99 and 134 min, respectively. In general, the BGFs with excellent fire resistance have great potential for fire protection applications.

## 1. Introduction

The term “Geopolymer” was first introduced by Joseph Davidovits in the late 1970s [1]. Geopolymer material is an amorphous aluminosilicate binder material. Geopolymers have been shown as a green alternative to ordinary Portland cement (OPC) due to their excellent mechanical properties, low permeability, low CO_2_ emissions, good chemical resistance, and excellent fire resistance [2]. Because of these advantageous properties, geopolymers have been used for making foam concretes, fire-resistant coatings, fiber-reinforced composites, etc. [3]. Among the products based on geopolymers, geopolymer foams (GFs) with highly porous structures have been considered as the most attractive materials over the past few years, because they possess exceptional properties, lightweight, low cost, and good fire and chemical resistances [4,5,6]. GFs can be prepared by several approaches including direct foaming, replica method, sacrificial filler method, and additive manufacturing [4]. Among these synthesis approaches, the direct foaming method is the most used technique for producing GFs using chemical blowing agents, such as hydrogen peroxide [7,8,9,10,11], metal powders [12,13,14,15], or silica fumes [16,17,18]. GFs have been used not only as sustainable building materials but also in a variety of applications, such as thermal and acoustic insulators [19,20,21], adsorbents and filters [7,22,23,24,25], and fire resistance [26,27,28,29].

Effects of elevated temperatures on properties of geopolymer materials have been extensively studied [30,31,32,33,34,35,36]. Geopolymers indicated a decrease in the mechanical strength when exposed to elevated temperatures [30,31,32,33]. Chithambaram et al. [31] reported that the compressive strength of geopolymer mortar reduced and its mass loss increased with increasing sample heating temperature to 1000 °C. Yang et al. [32] showed a reduction in the compressive strength and Young’s modulus of geopolymer made of red mud slurry and class F fly ash with the temperature rise to 1000 °C. Kürklü [33] indicated that coarse fly ash-based geopolymer mortars lost around 58% of strength at 1000 °C. However, several studies on GFs have shown that GFs exposed to high temperatures have good strength maintenance and low thermal shrinkage [34,35,36]. František et al. [34] reported that the mechanical properties of fly ash-based GFs were improved and stable at 1000 °C over the investigated period of a year. Hlaváček et al. [35] showed that the compressive strength of alkali-activated fly ash GFs improved significantly when samples heated at 1100 °C. Cilla et al. [36] indicated that the increase of treated temperatures caused a substantial enhancement in the compressive strength due to sintering and formation of crystalline phases.

Porous geopolymer materials have been widely considered in the field of fire protection due to their low density, low thermal conductivity, and excellent thermal insulation performance [26,27,28,29]. Sakkas et al. [26] showed that potassium based geopolymers retained its structure entirely after the fire test without any significant macroscopic damage and exhibited excellent fire resistant properties. Sarker et al. [27] examined the fire endurance of steel-reinforced fly ash geopolymer and OPC concrete plates. The geopolymer concrete elements showed greater fire resistance than the OPC counterparts. Peng et al. [28] reported that the amorphous skeleton structures of alkali-activated GFs have been converted to smooth ceramics after the treatment at high temperatures. Therefore, these GFs have a stable porous structure and extraordinary fire resistance. Shuai et al. [29] studied the fire resistance of phosphoric acid-based GFs and reported that the GFs possess quite low thermal conductivity and are useful for fire resistance applications. In addition, acid-based geopolymer materials could maintain structural stability and excellent mechanical properties at high temperatures [37,38]. The exceptional stability at high temperatures makes geopolymers as robust candidates in the development of GFs for fire protection.

GFs have been produced by a combination of geopolymer paste and fillers such as quartz sand, fly ash, silica fume, and foaming agents. Besides, several chopped fibers, e.g., organic fiber, glass fiber, basalt fiber, and mineral fiber have been used as the reinforcements in GFs to improve their mechanical strengths. Currently, GFs are excellent alternatives to several materials such as polystyrene, mineral wool, and glass, because they are a non-flammable material characterized by relatively good insulation [39]. Study of the effect of high temperatures over 1000 °C on the properties of GFs is essential to determine their fire resistance applicability. In this paper, basalt fiber-reinforced geopolymer foams (BGFs) made of ground basalt fibers, aluminum foaming agent, and potassium-activated metakaolin-based geopolymer have been produced. Effects of high temperatures to 1200 °C on the physical and mechanical properties of the BGFs, namely apparent density, water absorption, mass loss, drying shrinkage, and compressive and flexural strength were investigated. The BGFs were exposed to desired elevated temperatures for 2 h before measurements. The fire resistance of wood strand boards and steel plates coated by a protective BGF layer was examined through the fire resistance tests. The fire resistance test presented in this paper is not so commonly known. Changes in the porous structure, crystal structure, and chemical composition of BGFs after heat treatment at high temperatures and after fire resistance testing were presented. We explored potential use of potassium-activated BGFs for fire protection applications.

## 2. Materials and Methods

### 2.1. Materials

The industrially commercial geopolymer “Baucis lk” supplied by České Lupkové Závody, a.s. (Nové Strašecí, Czech Republic) is a two-component aluminosilicate binder based on metakaolin activated by alkaline solution of potassium. Basalt fibers were provided by Basaltex, a.s. (Šumperk, Czech Republic) and had a density of 2900 kg∙m^−3^ and thermal conductivity of 0.027÷0.033 W/m.K. The basalt fibers were ground using a grinding machine Bosch MW2514W (Robert Bosch GmbH, Stuttgart, Germany). The chemical compositions of the geopolymer Baucis lk and basalt fiber determined by X-ray fluorescence (BRUKER S8 Tiger instrument, BRUKER, Karlsruhe, Germany) used in making the BGF samples were analyzed and presented in Table 1. Round-shaped aluminum powders D50 with the mean size of 51.47 μm was supplied by Pkchemie Inc. (Třebíč, Czech Republic) were used to create pores inside the BGFs. The aluminum powder is 98% pure aluminum and has a chemical composition of 0.35% FeO, 0.4% SiO, and 0.02% Cu by weight. Photographs showing the materials for processing the BGFs are described in Figure 1.

### 2.2. Fabrication of BGFs

The BGFs were prepared by three following steps: (1) to begin with, a geopolymer mortar was prepared by mixing metakaolin-based geopolymer Baucis lk with an alkaline solution of potassium in a predetermined ratio (liquid to solid) by mechanical stirring; (2) afterwards, the ground basalt fibers were added to the geopolymer mortar mixture and the mixture was homogenized by the mechanical stirring; and (3) finally, aluminum powders were added to the mixture to create the BGF. Foaming of the mixture was carried out by the mechanical stirring. Three BGF samples with different mixing ratios and basalt fiber contents presented in Table 2 were prepared. All BGF samples for the measurement of mechanical and physical properties were cast in the mold and were cured for 28 days at room temperature (RT). The thermal conductivities of the samples S1, S2, and S3 were measured, respectively, as 0.137, 0.143, and 0.131 W/m·K [6].

### 2.3. Heat Treatment of the BGFs

To assess the properties of the BGFs at elevated temperatures, samples were exposed to elevated temperatures at 200, 400, 600, 800, 1000, and 1200 °C. The BGF specimens were heated in a furnace at a heating rate of 5 °C/min until the desired temperatures and were kept at each temperature in the furnace for 2 h. The heated BGF samples were then naturally cooled down to RT in the furnace. The heat treatment process of the BGF samples is presented in Figure 2.

### 2.4. Characterizations

The apparent density, water absorption, mass loss, and drying shrinkage of the unexposed and exposed BGFs to elevated temperatures were investigated. The BGF samples with the dimensions of 40 by 40 by 160 mm^3^ were used to measure the physical properties. Three specimens from each sample group were tested and their mean values were calculated for the physical properties of BGFs. The apparent density of BGFs was measured according to standard ČSN EN 1936 and was estimated by dividing the mass of the sample by its apparent volume.

Water absorption is used to measure the permeability of the BGFs. Water absorption of the BGFs was determined by the mass change between the dry and wet specimens using the standard ASTM C642-06. The dry samples were cured in an oven at the temperature of 100–110 °C for at least 24 h. The wet samples were soaked in the water for a 24 h interval. The water absorption was calculated using the following equation:(1)$$Wa\;\%=\frac{A-A_{0}}{A_{0}}\times 100\%$$
where Wa is the percentage of water absorption, *A_0_* is the mass of the dry sample (g), and *A* is the mass of the wet sample (g).

The drying shrinkage of the BGFs was determined by measuring the length of the BGF samples before and after exposure to elevated temperatures. The difference in length of samples before and after exposure to elevated temperatures indicates the drying shrinkage. Similar method was also used to determine the mass loss of BGFs after exposure to elevated temperatures.

A scanning electron microscope (SEM) ZEISS Ultra Plus equipped with EDX detector (ZEISS, Oberkochen, Germany) was used to investigate microstructural morphology of the unexposed and exposed BGFs to elevated temperatures. X-ray diffraction (XRD) analysis was performed to examine the phase composition and the crystalline content of metakaolin-based geopolymer and basalt fibers at different temperatures. XRD patterns were recorded using a Bruker D8 Advance XRD system equipped with a Bruker SSD 160 detector and operating with Cu-Kα radiation at 40 kV and 25 mA.

### 2.5. Mechanical Testing

Mechanical testing was conducted to measure mechanical strengths of the unexposed and exposed BGFs to elevated temperatures. Compressive strength of the BGFs was measured using 40 mm^3^ cubic specimens. Flexural strength was measured from a three-point bending test for the samples with a span length of 100 mm and the dimensions of 40 by 40 by 160 mm^3^. The compressive and bending tests were conducted with a load cell of 10 kN at a crosshead speed of 2.0 mm/min at laboratory temperature of about 22 ± 3 °C using a universal testing machine Instron (Model 4202). Mean values of compressive and flexural strengths were obtained from three specimens for each series.

### 2.6. Fire Resistance Test

To evaluate the fire protection of the BGFs, the fire resistance tests were conducted on the wood strand boards and steel plates coated with a BGF layer type S3. Fire resistance specimens were fabricated by coating the BGF mixture type S3 on the surface of wood and steel plates by casting and were cured for 28 days at RT before testing. The thicknesses of wood and steel boards before coating the BGF layer are 22 and 2 mm, respectively. The fire resistance samples have a 2D dimension of 500 by 500 mm^2^. The fire exposure region of the samples was 300 by 300 mm^2^. The thicknesses of the coated BGF layers for the fire resistance specimens are presented in Table 3.

The inside, outside and chimney temperatures of the fire test furnace were measured using in-built thermocouples connected to the computer by ADAM 4000 series. The respective thermocouples T1, T2, and T3 were mounted on the fire-exposed surface of the specimen, unexposed sample surface, and in the furnace chimney. The fire test furnace, position of thermocouples, the ADAM 4000 series system, laptop, and others were depicted in Figure 3.

The test furnace was heated by a system of natural gas. The fire resistance test for the wood specimens was stopped once shiny spots began to appear on the non-fire surface of samples and a little smoke escaped. The endpoint of the fire resistance test for the steel specimens was done when observing the deformation (e.g., buckling and warping defects) of the unexposed surface or the measured temperature on the unexposed surface of samples reached 300 °C using an infrared thermometer Voltcraft IR 650-16D. The furnace fire was controlled to obtain the heating rate recommended in the standard ISO 834 [40]. The furnace temperature was determined by the following equation:(2)$$T_{t}=T_{0}+345\log_{10}\left(8t+1\right)$$
where *T*_t_ is the furnace temperature (°C) at time *t* (min) and *T*_0_ is the initial temperature (°C).

## 3. Results and Discussion

### 3.1. Temperature-Dependent Properties

The physical and mechanical properties of the unexposed and exposed BGFs to elevated temperatures were measured. The apparent density, water absorption, mass loss, and drying shrinkage as functions of temperature for the BGFs are shown in Figure 4. Effects of high temperatures on the compressive and flexural strengths of the BGFs are presented in Figure 5. Photographs describing the structural morphologies of unexposed and exposed BGFs to elevated temperatures are given in Figure 6. Optical photographs illustrating the microstructural morphologies of untreated and heat-treated BGF specimens are presented in Figure 7. SEM micrographs of unexposed and fire-exposed BGF samples type S3 at a number of specific temperatures are portrayed in Figure 8. XRD patterns of the metakaolin-based geopolymer and basalt fiber after heat treatment at different elevated temperatures are shown in Figure 9. The chemical element compositions of the unexposed and exposed BGF sample S3 to elevated temperatures which were determined using SEM equipped with EDX detector are given in Table 4.

As observed in Figure 4a, the apparent density of the BGFs decreases slightly with increasing the treated temperature to 400 °C, then enhances as the temperature increases to 600 °C. The apparent density does not vary significantly at temperatures between 600 and 800 °C, but increases rapidly with the rise of the temperature from 800 to 1200 °C. The respective densities of the S1, S2, and S3 after heat treatment at 1200 °C were 737, 717.1, and 804.8 kg/m^3^ and they increased by 23.2%, 14.4%, and 21.4% compared to those at RT. Figure 4d showed that the drying shrinkage of the BGFs changes slightly with the rise of treated temperatures until 400 °C, then enhances sharply over 400 °C. The drying shrinkages of the S1, S2, and S3 at a temperature of 200 and 400 °C were only approximately 1%, but they respectively increased to 11.4%, 9.8%, and 8.5% at 1200 °C. Besides, when increasing the basalt fiber content, the drying shrinkage of the BGFs decreases. Moreover, the drying shrinkages of unexposed and exposed samples to elevated temperatures can be seen in Figure 6.

The water absorptions of the BGFs are presented in Figure 4b. As Figure 4b shows, the highest water absorptions of the S1, S2, and S3 occurred at 200 °C were 62.3%, 61.5%, and 55.4%, respectively. The water absorption of the S1 decreases drastically with increasing the temperature from 200 to 800 °C, then gradually increased as the temperature rises to 1200 °C. Similarly, the water absorption of the S2 dropped rapidly with the increase of the temperature from 200 to 600 °C, next changed moderately to 1000 °C, and finally increased sharply to about 50% at 1200 °C. The water absorption of the sample S3 fell remarkably as the temperature rose to 400 °C and then fluctuates at around 35% beyond 400 °C. It is interesting that the S1, S2, and S3 showed the similar water absorption at a temperature of 1000 °C. Particularly, the water absorption of the sample S3 with higher basalt fiber content was lower than that of samples S1 and S2 at temperatures of below 600 °C and above 1000 °C.

The mass losses of the BGFs are shown in Figure 4c. The mass losses of the S1, S2, and S3 occurred at 200 °C, respectively, were 14.9%, 14.0%, and 5.0%. Below 600 °C the mass loss of S3 was just about one third compared to that of the S1 and S2. The mass loss of the S1 and S2 raised with increasing the temperature to 600 °C, then remained almost unchanged at temperatures between 600 and 1000 °C, and finally declined rapidly to about 10% at 1200 °C. The mass loss of BGFs heated at 1200 °C was lower than that heated at 1000 °C, because the BGF samples exposed to 1200 °C absorb more water during cooling than those exposed to 1000 °C. For the sample S3 with higher basalt fiber content, the variation of the mass loss below 1000 °C was the same as that of the S1 and S2. The mass loss of the S3 is almost constant at temperatures between 600 and 1000 °C, then reduced slightly to about 10% at a temperature of 1200 °C. Maximum mass losses for the S1, S2, and S3 exposed to 1000 °C were found to be 22%, 19.2%, and 10.6%, respectively. At 1200 °C the mass losses of the S1, S2, and S3 were approximately the same. It is interesting that the mass loss of the S3 was much lower than that of samples S1 and S2 at temperatures under 1000 °C. In general, below 1000 °C the mass loss decreases drastically as the basalt fiber content in the BGFs increases.

The compressive and flexural strengths of the BGFs are presented in Figure 5. As observed in Figure 5a, the compressive strength of the BGFs changed somewhat at temperatures below 200 °C and then decreased slightly in the temperature range from 200 to 400 °C. Once the temperature increased from 400 to 800 °C, the S1 showed a slight reduction in the compressive strength, while the S2 indicated an enhancement in the compressive strength. Unlike the S1 and S2, the S3 showed that the compressive strength increased with increasing the temperature to 600 °C and then remained constant between 600 and 800 °C. The compressive strengths of the S1, S2, and S3 enhance with the rise of the temperature from 800 to 1000 °C. As the temperature increased from 1000 to 1200 °C, the S1 and S3 showed an increase in the compressive strength, whereas the S2 indicates an inconsiderable reduction. Interestingly, the compressive strength of the BGFs heat-treated at 1200 °C was improved significantly compared with that at temperatures below 200 °C. The respective samples S1, S2, and S3 after heating at 1200 °C showed an increase in the mean compressive strength by 48.0%, 39.1%, and 107.6% compared to those at RT. In addition, the compressive strength enhanced with raising the basalt fiber content.

xSimilarly to the compressive strength, the flexural strength of the BGFs varied slightly at temperatures under 200 °C and then decreased at temperatures between 200 to 400 °C (see Figure 5b). A sharp reduction in the flexural strength occured for samples with low basalt fiber contents (S1 and S2). When the temperature rose from 400 to 800 °C, the variation in the flexural strength of the S1, S2, and S3 is similar to that of their compressive strength. Although the S2 and S3 showed an increase in the flexural strength at temperatures between 800 to 1000 °C, the S1 indicated the unchanged flexural strength. All three samples showed a rapid increase in the flexural strength as the temperature enhanced from 1000 to 1200 °C. However, the sample S1 at 1200 °C showed a lower flexural strength compared with that at RT, whereas the respective samples of S2 and S3 at 1200 °C indicated a higher flexural strength by 53.6% and 115.9% compared to those at RT. It is clear from Figure 5b that the flexural strength enhanced with the rise of basalt fiber content. Moreover, the flexural strengths of the BGF with highest fiber content (S3) was improved incredibly at high temperatures.

It is interesting that the compressive and flexural strengths of the BGF with highest fiber content (S3) were improved considerably at high temperatures over 600 °C. Unlike the S3, the S1 with low fiber loading showed a decrease in its compressive and flexural strengths at temperatures below 1000 °C. This reduction is attributable to macroscopic cracks appeared in samples (Figure 6b). However, the cracks did not appear in the BGF samples with the high fiber content (Figure 6c,d). The strength increase of the S3 was due to reinforcing the basalt fibers in the metakaolin-based geopolymer, because basalt fibers showed a good strength and stiffness at high temperatures [41]. In addition, all three samples indicate a great enhancement in the compressive and flexural strengths as the temperature rose from 1000 to 1200 °C. The improvement in the mechanical strengths of the BGFs at high temperatures is ascribable to their structural changes, including sintering, densification, melting, and porosity [4].

The occurrence of sintering, densification and melting makes decolorization in geopolymer mortars when the temperature over 1000 °C [32]. Figure 7 showed the decolorization and fiber appearance of the BGF samples when heating them at high temperatures. As observed in Figure 7, the BGF samples have a similar color at temperatures below 200 °C. The color of the BGF samples changed slightly in the temperature range from 400 to 1000 °C. The BGF sample at 1200 °C looked brighter than the others at lower temperatures. In addition, basalt fibers are visible at temperatures below 1000 °C and were not observed at 1200 °C. Ye et al. [32] reported that the increase in the compressive and flexural strengths of geopolymer mortars is attributable to melting and densification effects after heating at 1000 °C. In our study, the melting of potassium-based geopolymer matrix was observed after the BGFs exposed to high temperatures over 1000 °C (Figure 8b). The melting of the geopolymer matrix caused the densification of the BGFs (Figure 7u and Figure 8c) and facilitated a strength increase at 1200 °C (Figure 5).

Moreover, the formation of ceramic phases in the BGFs at 1200 °C (Figure 8) enhanced, considerably, their compressive and flexural strengths (Figure 5). The phase changes of the metakaolin-based geopolymer and basalt fibers at high temperatures are depicted in Figure 9. Evidently, there was the presence of amorphous phases in the metakaolin-based geopolymer at RT [28,36,42]. The XRD patterns in Figure 9a exhibited no significant phase change in the geopolymer samples below 800 °C. However, over 1000 °C exposure the amorphous phases converted into crystalline phases due to geopolymerization mechanism, such as leucite (KAlSi_2_O_6_), akermanite (Ca_2_MgSi_2_O_7_), anorthite (CaAl_2_Si_2_O_8_), and diopside (CaMgSi_2_O_6_). These phases are stable crystalline phases and have a great stability at high temperatures. The strong peaks at 2θ angles of 27° and 30° in the geopolymer mainly identify the presence of leucite and akermanite phases at 1000 and 1200 °C. The leucite peaks at 1200 °C are higher than those at 1000 °C.

The XRD patterns in Figure 9b also showed that no remarkable phase change has taken on the basalt fiber exposed to 600 °C, similarly to the metakaolin-based geopolymer (Figure 9a). Nevertheless, crystalline phases as akermanite, phonotephrite, and nepheline start to form at 800 °C and become stable around 1000 °C. The phonotephrite phase appeared in the basalt fiber peaks at 2θ angles of 30° and 35°. Bayrak and Yilmaz reported that the akermanite and nepheline phases were identified in granite-based glass-ceramic materials at about 1000 °C [43]. The appearance of the crystalline phases proved that the metakaolin-based geopolymer reinforced with basalt fibers exposed to temperatures over 1000 °C transformed into a ceramic material with higher strength and stiffness. As seen in Figure 8c, there were crystalline phases and densification in the microstructure of the BGFs. Therefore, the heat treatment of BGFs at high temperatures created a denser crystalline structure, resulting in a considerable increase in their mechanical strengths (Figure 5). In general, the BGFs have the low thermal conductivity and excellent mechanical stability at high temperatures. As a result, they can be appropriate for fire protection applications.

### 3.2. Fire Resistance of BGFs

Fire resistance testing was conducted on the wood and steel plates coated by a BGF layer type S3 with different thicknesses using the flame of a gas burner. The samples of the wood boards and steel plates for fire resistance test were presented in Table 3. The temperature-time curves of the fire resistance test for the wood samples are presented in Figure 10. The fire-resistant times for the samples WS1, WS2, and WS3 were 22, 49, and 99 min, respectively. The fire-resistant time increased with increasing the thickness of coated BGF layer. The fire-resistant time of the wood sample with 20 mm thickness of coated BGF layer was 4.5 times higher than that without coating the BGF layer. The temperatures on the fire-exposed surface of all samples increased rapidly during the initial period and then enhanced slowly beyond 15 min. The temperatures on the unexposed surface of all samples raised gradually during the fire resistance testing. The non-fire surface temperatures of the wood samples did not exceed 200 °C, while the maximal temperature on their fire-exposed surface did not pass 800 °C.

As with the wood boards, the fire resistance of the steel samples which was shown in Figure 11 depended on the thickness of the coated BGF layer. The fire-resistant time of the steel plates covered with the BGF layer was 41 min for the SS2 and 134 min for the SS3, but it was just 9 min for the SS1 (steel plate without coating) in the same testing condition. The fire-resistant time of the SS3 was about 15 times higher than that of the steel plate without coating the BGF (SS1). The maximal temperature on the fire-exposed surface of samples in the furnace did not exceed 900 °C and the unexposed surface temperatures did not surpass 300 °C. It is noted that the fire-resistant time of the wood and steel plates coated by a BGF layer depends not only on the coating thickness but also on the thermal conductivity of the substrate. With the same BGF coating thickness, the fire-resistant time of the steel sample (SS3) was 2.73 times higher than that of the wood sample (WS2).

It is clear from Figure 12 that the surfaces of the wood and steel specimens coated by the BGF did not change significantly before and after fire resistance testing. The BGF retained its structure entirely without any significant macroscopic damage after fire resistance test, as reported by Sakkas et al. [26]. Macroscopic cracks did not appear on the exposed surfaces of the samples, but there was a change in the color of their surfaces after the fire resistance test. The slight color variation of the sample surfaces after the fire resistance testing is attributable to the fire-exposed surface temperature in the furnace lower than 900 °C. This temperature did not affect considerably the surface structure of the BGFs with high fiber content. As presented above, the BGFs exposed to high temperatures will become stronger. Therefore, the BGFs can be used as excellent coating materials for fire resistance applications.

## 4. Conclusions

The BGFs based on ground basalt fibers, aluminum foaming agent, and potassium-activated metakaolin-based geopolymer have been developed. Temperature-dependent properties of the BGFs, including apparent density, water absorption, mass loss, drying shrinkage, and compressive and flexural strengths, were measured and assessed. The following conclusions were pointed out:The apparent density and drying shrinkage of the BGFs increased with increasing the temperature from 400 to 1200 °C. Under 600 °C the mass loss was enhanced while the water absorption was reduced and they both changed slightly between 600 and 1000 °C. Over 1000 °C the mass loss was decreased considerably, whereas the water absorption was enhanced. The compressive and flexural strengths of the high fiber loading BGFs were improved significantly at temperatures above 600 °C and reached the maximum at 1200 °C.The BGFs exposed to high temperatures exhibited a smooth change from amorphous to crystalline phase. Moreover, the BGFs with high fiber content exposed to high temperatures resulted in the dense crystalline structure, thereby improving their mechanical strengths. Therefore, the high fiber content BGFs showed excellent mechanical stability at high temperatures.The fire resistance of the wood and steel plates has been considerably improved after coating a BGF layer on their surface. The coated BGF kept its structural integrity without any substantial macroscopic fracture after fire resistance test. The longest fire-resistant times for the wood and steel boards were 99 and 134 min, respectively.

In short, the BGFs with extraordinary fire resistance have great potential for fire protection applications.

## Figures and Tables

**Figure 1 polymers-12-02994-f001:**
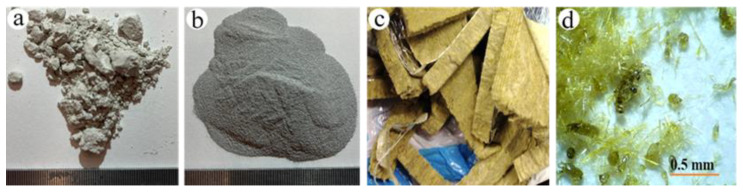
Photographs showing the materials: (**a**) geopolymer Baucis lk, (**b**) aluminum powders, (**c**) basalt fibers before grinding, and (**d**) basalt fibers after grinding.

**Figure 2 polymers-12-02994-f002:**
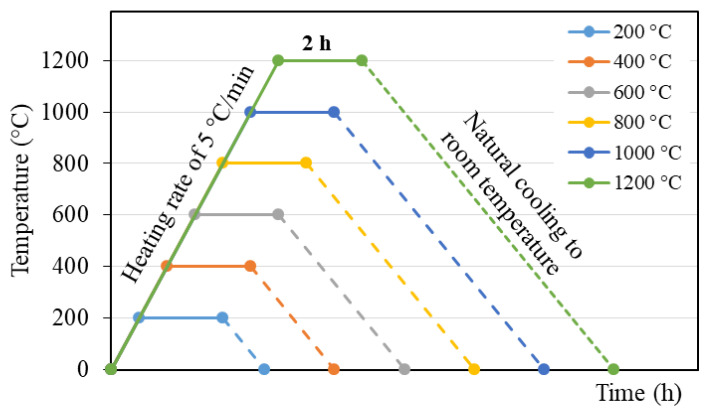
Heat treatment process of the BGF samples.

**Figure 3 polymers-12-02994-f003:**
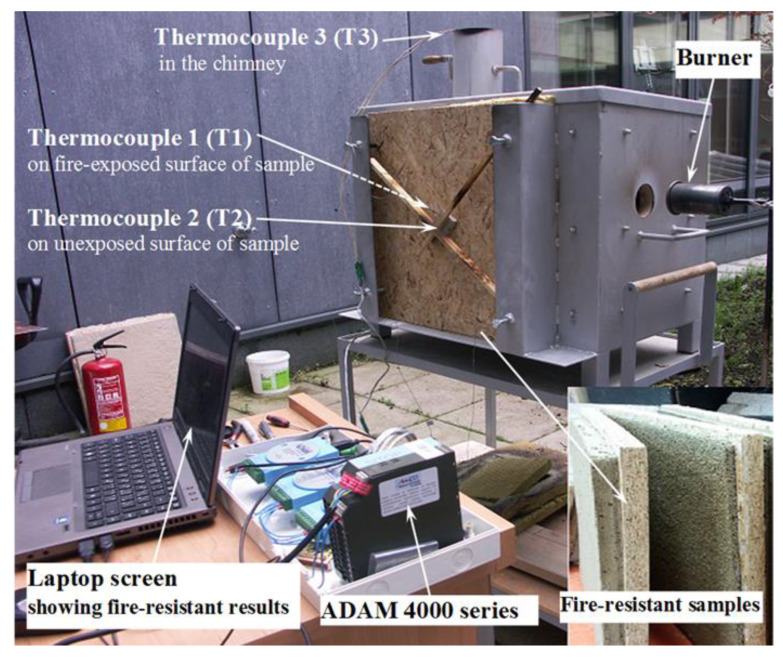
An image illustrating equipment for fire resistance test.

**Figure 4 polymers-12-02994-f004:**
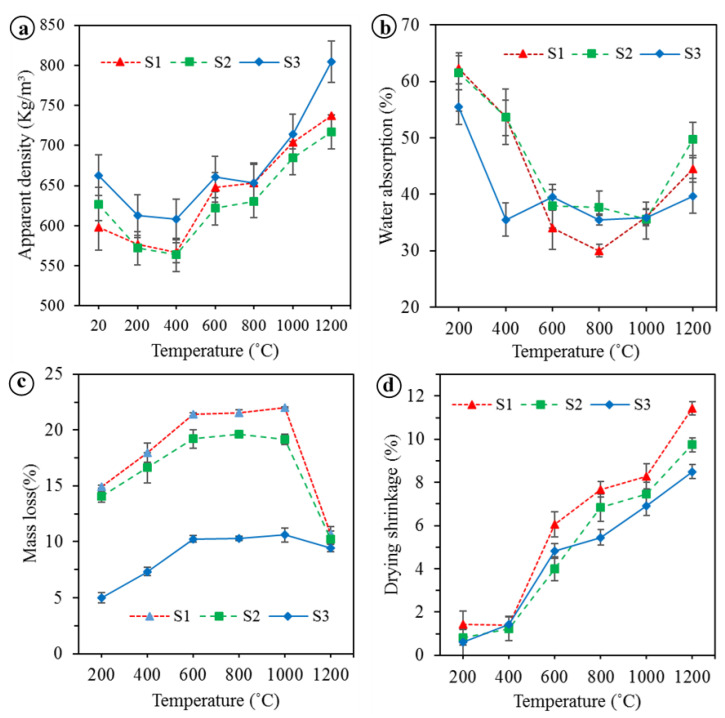
The physical properties of the BGFs at different temperatures: (**a**) apparent density, (**b**) water absorption, (**c**) mass loss, and (**d**) drying shrinkage.

**Figure 5 polymers-12-02994-f005:**
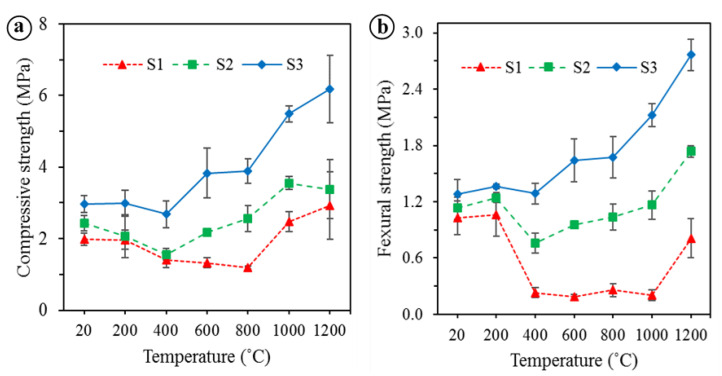
(**a**) Compressive strength and (**b**) flexural strength of the BGFs at different temperatures

**Figure 6 polymers-12-02994-f006:**
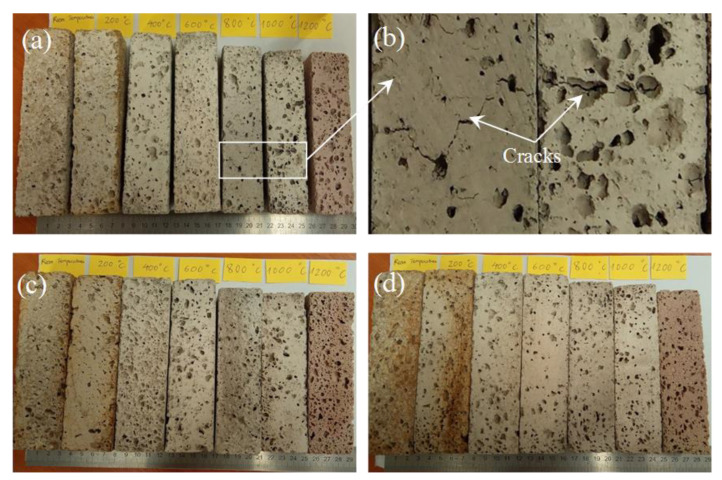
Photographs illustrating structural morphologies of different BGF samples after heat treatment at elevated temperatures: (**a**) and (**b**) S1, (**c**) S2, and (**d**) S3.

**Figure 7 polymers-12-02994-f007:**
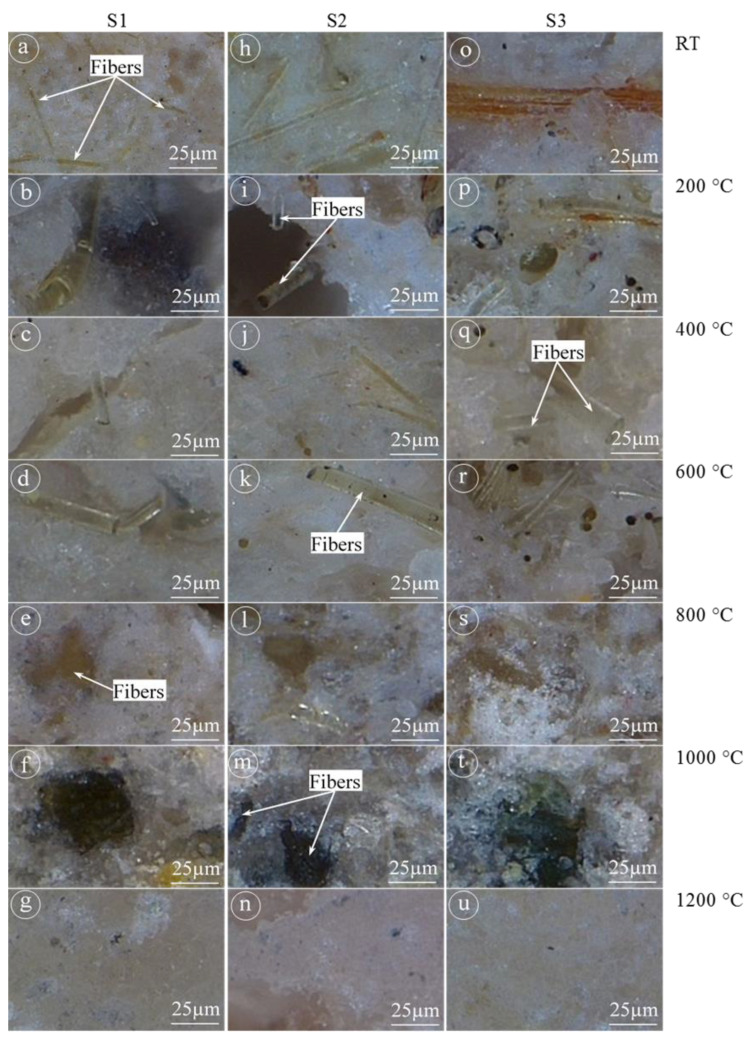
Optical micrographs of BGF samples at elevated temperatures: (**a**–**g**) for S1, (**h**–**n**) for S2, and (**o**–**u**) for S3.

**Figure 8 polymers-12-02994-f008:**
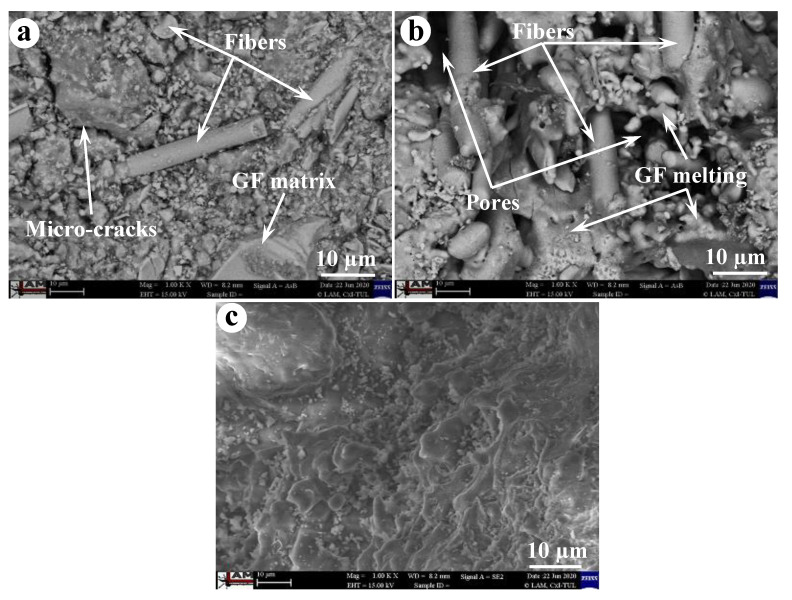
SEM micrographs of BGF samples type S3 at several specific temperatures: (**a**) room temperature (RT), (**b**) 1000, and (**c**) 1200 °C.

**Figure 9 polymers-12-02994-f009:**
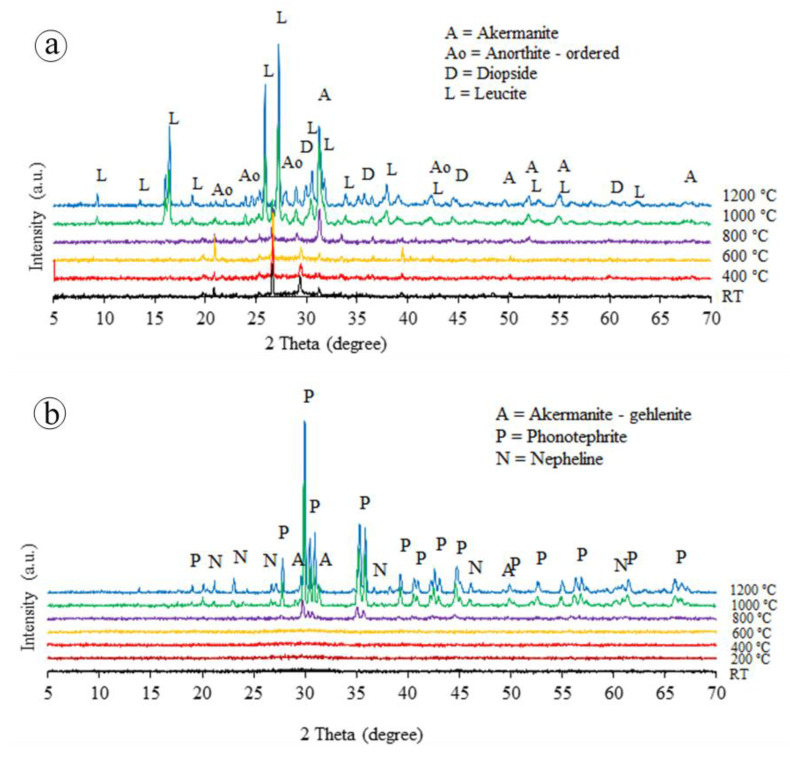
XRD patterns of (**a**) metakaolin-based geopolymer and (**b**) basalt fiber at different elevated temperatures.

**Figure 10 polymers-12-02994-f010:**
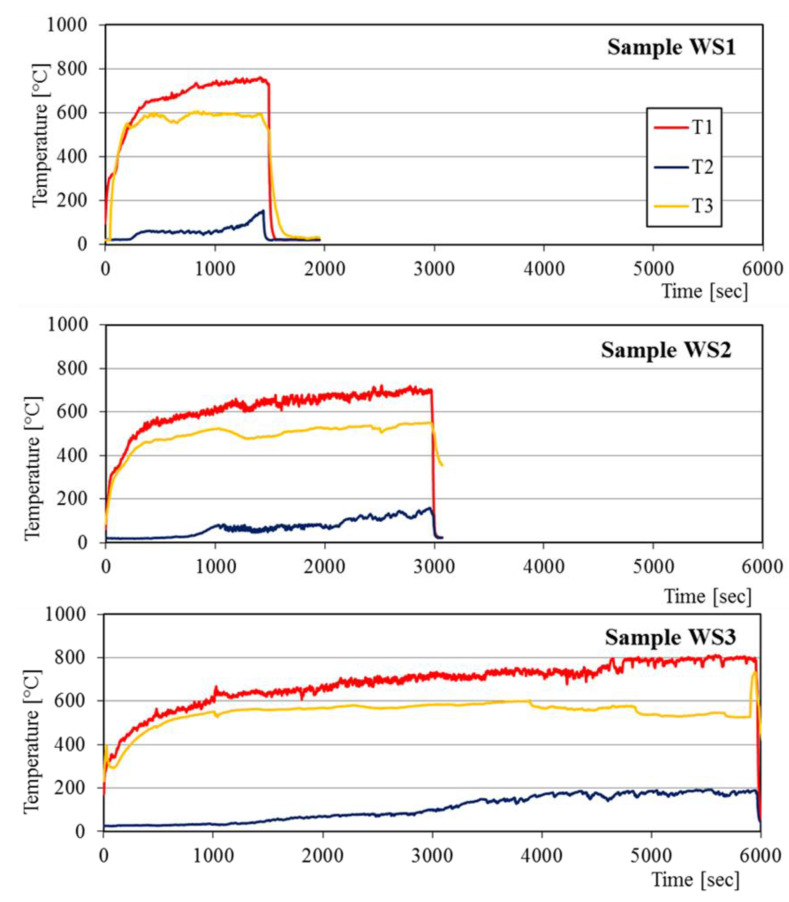
Temperature-time curves of fire resistance tests for the wood samples.

**Figure 11 polymers-12-02994-f011:**
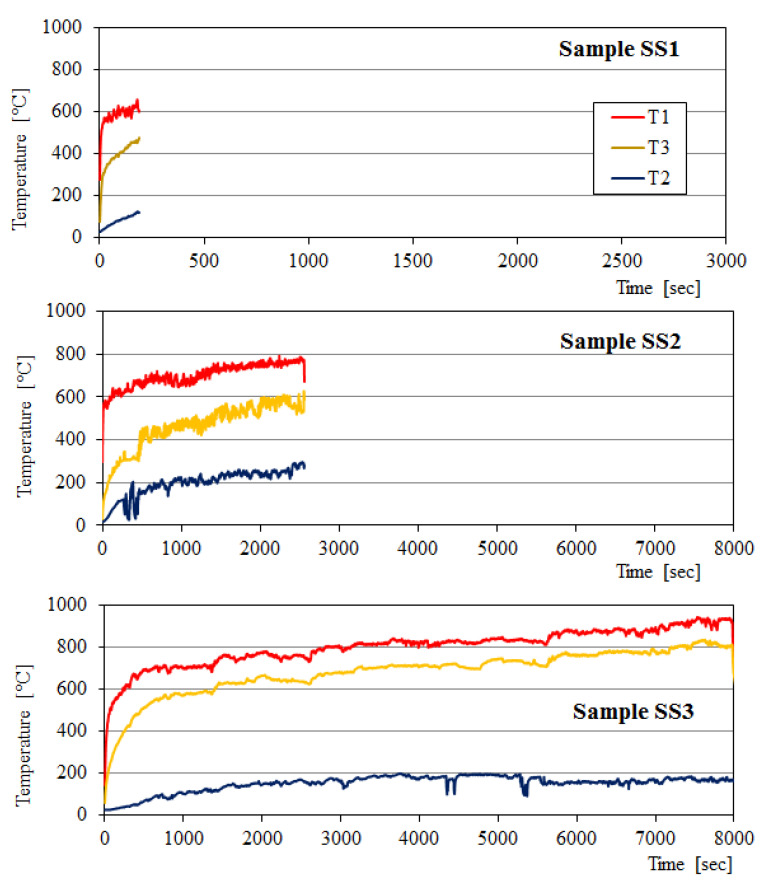
Temperature-time curves of fire resistance tests for the steel samples.

**Figure 12 polymers-12-02994-f012:**
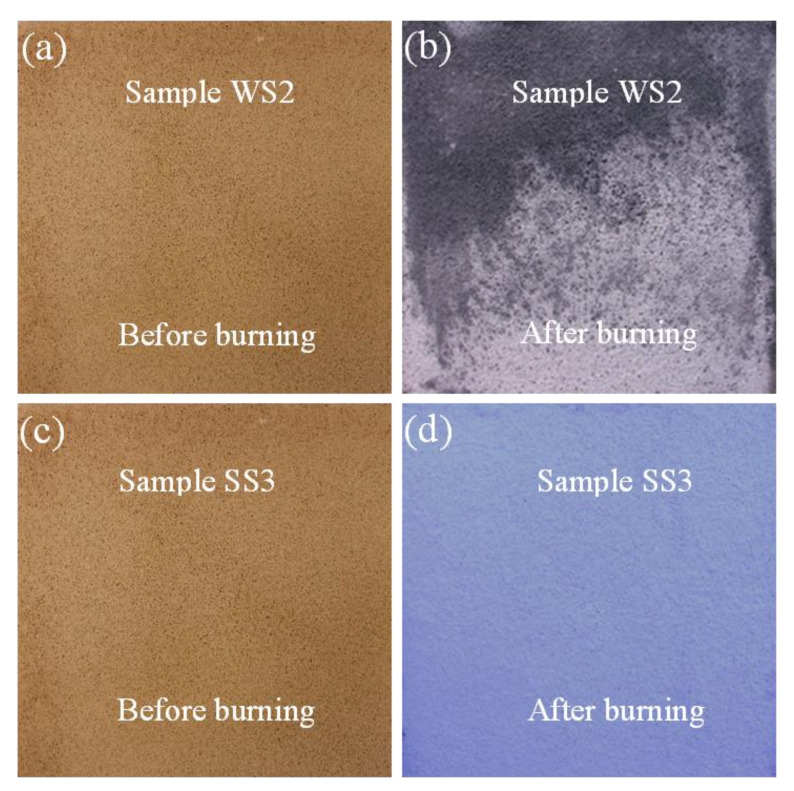
Photographs showing the fire-exposed surfaces of the samples WS2 and SS3: (**a**) and (**c**) before burning, (**b**) and (**d**) after burning.

**Table 1 polymers-12-02994-t001:** Chemical compositions of geopolymer Baucis lk and basalt fiber (wt.%).

Constituents	SiO_2_	Al_2_O_3_	CaO	MgO	TiO_2_	Fe_2_O_3_	K_2_O	SO_3_	MnO	Na_2_O	P_2_O_5_	LOI
Geopolymer	44.5	28.9	17.6	2.23	1.31	0.82	0.75	0.46	0.28	0.25	−	2.56
Basalt fiber	33.6	14.4	26.1	8.26	1.98	6.61	1.21	0.29	0.76	1.38	0.14	2.05

**Table 2 polymers-12-02994-t002:** Mix proportions of basalt fiber-reinforced geopolymer foam (BGF) samples by weight ratio.

BGF Sample	Binder		Aluminum Powder/Binder	Basalt Fiber/Binder
	Baucis lk	Activator		
S1	1	0.9	0.008	0.05
S2			0.008	0.16
S3			0.008	0.26

**Table 3 polymers-12-02994-t003:** Wood and steel samples for fire resistance test.

Base Material	Wood			Steel		
Sample ID	WS1	WS2	WS3	SS1	SS2	SS3
Thickness of the coated BGF layer (mm)	0	10	20	0	5	10

**Table 4 polymers-12-02994-t004:** Chemical element composition of the sample S3 at different temperatures (wt.%).

Constituents	C	O	Na	Mg	Al	Si	P	S	Cl	K	Ca	Ti	Mn	Fe	Total
RT	5.59	45.76	0.20	1.05	10.23	17.91	0.00	0.06	0.11	10.0	7.91	0.42	0.21	0.56	100
1000 °C	4.07	43.53	0.29	2.46	9.60	18.60	0.07	0.12	0.00	6.72	11.65	0.66	0.50	1.72	100
1200 °C	4.77	43.96	0.29	1.55	10.52	21.48	0.03	0.05	0.00	7.82	8.24	0.40	0.19	0.71	100

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
