# Peer review of "Study on Temperature-Dependent Properties and Fire Resistance of Metakaolin-Based Geopolymer Foams"

_polymers, 2020, doi:10.3390/polym12122994_

Round 1

Reviewer 1 Report

The article under the title: “Study on temperature-dependent properties and fire resistance of metakaolin-based geopolymer foams” is appropriate for Polymers journal. The authors present interesting and up-to-date topic connected with materials based on inorganic polymers composites, so called geopolymers. The organization of the article is typical for research article. Overall, the paper is well prepared, but it requires some improvements:

- Introduction (line: 36-37): “Geopolymer which was first introduced by Joseph Davidovits in the late 1970s is an amorphous aluminosilicate binder material”, please be more specific the term “geopolymer” was introduced by J. Davidovits not the material itself.

- Introduction (line: 39): “CO2” please apply down index.

- Introduction: please stress the novelty of presented research comparison with analysed literature.

- Materials and Methods (line 98-100): “The chemical 98 compositions of the geopolymer Baucis lk and basalt fiber used in making the BGF samples were 99 analyzed and presented in Table 1.” – please give information about the method of this analysis.

- Materials and Methods: please verify the date in Table 2 (binder).

- Discussion: lack of discussion with the literature (IMPORTANT); Please compare received results with up-to-date-literature, for example:

  • Fire Resistance of Alkali Activated Geopolymer Foams Produced from Metakaolin and Na2O2 / Xi Peng, Han Li,Qin Shuai, Liancong Wang // Materials – 2020, Vol. 13(3), 535
  • Development and characterization of thermal insulation geopolymer foams based on fly ash / Michał Łach, Janusz Mikuła, Wei-Ting Lin, Patrycja Bazan, Beata Figiela, Kinga Korniejenko // Proceedings of Engineering and Technology Innovation – 2020, Vol. 16, p. 23-29
  • Thermal phenomena of alkali-activated metakaolin studied with a negative temperature coefficient system / Dariusz Mierzwiński, Michał Łach, Marek Hebda, Janusz Walter, Magdalena Szechyńska-Hebda, Janusz Mikuła // Journal of Thermal Analysis and Calorimetry – 2019, Vol. 138, Iss. 6, p. 4167-4175.
  • And other, including references in the article.

- Figure 12: please change the localization.

- Author Contributions: Lack of proper information.

Author Response

Dear reviewer, Thank you very much for your valuable comments. I will response to the comments made by the reviewer. I hope I can comply with the requirements in a sufficient way to make the manuscript publishable. In addition to the changes made in the revised manuscript, I will answer point by point the statements of the reviewer.

Yours sincerely,

Van Su Le

Reviewer 2 Report

The manuscript present new and refine geopolymer with multiple functional properties. Here are some comments:

  1. I suggest the authors to define the need of such geopolymers, the significance and related works should be added in the introduction part.
  2. The references are outdated, I’m sure there must be many related works in literature.
  3. Have the authors think of high temperature testing, e.g., Pyrolysis-combustion flow calorimetry. Or perhaps the evolved gas during the combustion of such systems? It may generate toxins.
  4. The results should be systematically presented, add Fig. 12 in the result and discussion part. Moreover, conclusion should be exclusive and conclusive, mentioning major findings. Most importantly add a line about their potential uses.
  5. The manuscript needs language revisions, mostly sentences are not formulated well.

Author Response

(The authors gave the same response as above.)

Reviewer 3 Report

Dear authors,

The article presents a large range of results. It is interesting and deserves to be accepted for publication.  Nevertheless, there is a couple of issues to be addressed before publishing it.

Three formulations have been studied with different basalt fiber loading. Unfortunately there is no formulation without basalt fiber. Why didn’t you prepare such formulation?

You choose 300 °C as the critical temperature to assess the fire resistance. Why did you choose such temperature? For building, the criterion I (thermal insulation) fails when the temperature of the unexposed surface reaches 140°C (or 180 °C).

The first part of the section Results describes the results. In the second part, based on XRD analyses, we understand that these results are assigned to water release and then sintering/crystallization at higher temperature. Neverthless, some results are unclear or not well explained. First, Figure 4a shows that the lower density is obtained for S2 (i.e. for the intermediate basalt loading). Similarly, water absorption is the highest for this formulation (at 1200°C). Have you an explanation about that?

I do not understand why mass loss decreases between 1000 and 1200 °C (Figure 4c). I suppose mass loss is calculated between mass at a fixed temperature and the initial mass. Then how can the mass loss decrease between 1000 and 1200 °C? Please clarify.

Be careful about some presentations of results. For example, lines 249-250, you write that “the respective samples at 1200°C showed an increase in the compressive strength.” In fact a more correct sentence is that “the respective samples showed an increase in the compressive strength after heating at 1200°C” (and cooling – because I suppose that the mechanical performances are measured after cooling).

I do not understand some explanations lines 288-294. You write that “BGFs exposed to high temperatures accelerate the melting of geopolymer matrix because the geopolymer binder in the BGFs begins to melt at about 1000°C”. But how can you prove that BGFs accelerate the melting. And which would the mechanism be?

Lines 299-306, you explain that the ratio Si/Al increases at high temperature. This increase would explain why mechanical properties increase. But how can this ratio increase? It is unclear for me. Please clarify. Moreover, the sintering and the resulting density increase can explain the improvement of mechanical properties.

In some parts, you provide some explanations based on thermal conductivity (for example lines 329-330). But you did not measure the thermal conductivity of the formulations. Can you measure it? If not, I think this hypothesis (high fiber content have a low thermal conductivity) should be removed. Indeed, thermal conductivity is probably more driven by density.

I am not sure to understand where is located the thermocouple on the exposed surface? Is it located on the exposed surface of the coating or between the coating and the wood/steel substrate? If it is located directly on the exposed surface of coating, the temperature should be the same in all cases and should correspond to ISO 834 standardized fire. Please clarify.

In some graphs (for example Figure 11 – SS1), the curve of T2 does not reach 300°C. There how did you assess the fire resistance (because you consider that the endpoint of the test is reached when the temperature of the unexposed surface reaches 300°C) ?

The sentences in lines 357-360 is unclear. Please explain more clearly.

Sincerely

Author Response

(The authors gave the same response as above.)

Round 2

Reviewer 2 Report

The authors have put efforts to revise the manuscript, it would be better if they add some sentences on the consumption of fire resistance in different fields.

Add the following paper for some stats of fire resistance (Global consumption of flame retardants and related environmental concerns: A study on possible mechanical recycling of flame retardant textiles).

And, this on how they work (Process optimization of eco-friendly flame retardant finish for cotton fabric: A response surface methodology approach).

Author Response

Dear reviewer,

Thank you very much for your comment. The manuscript has presented the effect of high temperature on physical and mechanical properties of geopolymer foams and their fire resistance as well. Therefore, the manuscript authors only referred to the papers closely related to research topics (temperature-dependent properties and fire resistance of geopolymer materials). We deeply appreciate the reviewers' understanding.

Yours sincerely

Le Van Su

Reviewer 3 Report

Dear authors,

You have addressed properly most of the issues pointed out by the reviewers. But I am still not convinced by 2 answers.

  • About mass loss, the mass loss is lower when BGF is heated at 1200 °C than at 1000 °C. But before reaching 1200°C, the sample reaches 1000°C. Then I do not understand how they can “gain” mass between 1000 and 1200°C. Maybe the water uptake during cooling is higher after heating at 1200 °C? Please clarify.
  • About the ratio Si/Al : I know that the ratio Si/Al influences the mechanical properties. But I do not understand why this ratio should change during heating. I suppose that the lost mass is mainly water. Then, the amount of Si and Al should be recalculated and the ratio Si/Al may change slightly. But I do not understand how the content of Al can decrease while the content of Si increases (as at 1000 °C for S3). The element analysis is carried out on the whole material, isn’t it ?
  • Last point about this sentence “Currently, GFs are excellent alternatives to several materials such as polystyrene, mineral wool, and glass, because they are a non-flammable material characterized by relatively good insulation [39].” I agree but the density of GF remains much hgher than polystyrene foams. I am not sure that both materials are suitable for the same applications.

Sincerely

Author Response

Dear reviewer

Thank you very much for your valuable comment. I responded to your comments in the cover letter in a sufficient way to make the manuscript publishable in Polymers.

Yours sincerely,

Le Van Su

Round 3

Reviewer 3 Report

Dear authors,

I am satisfied by your answers. But you must change the text accordingly:

  • If the decrease in mass loss at 1200°C is explained by water absorption, the text must contain this explanation.
  • If the element analysis is not really reliable because it is done on a very small surface area, the paragraph strating in line 312 must be removed.

Except if I am wrong, but the manuscript has not been revised.

Best regards

Author Response

Dear reviewer

Thank you very much for your valuable comments. I have put efforts to respond to your comments in a sufficient way to make the manuscript publishable in Polymers.

Yours sincerely,

Le Van Su
